# Effectiveness of Dengue Awareness Calendar on Indigenous Population: Impact on Knowledge, Belief and Practice

**DOI:** 10.3390/healthcare11050637

**Published:** 2023-02-21

**Authors:** Li Ping Wong, Arulvani Rajandra, Juraina Abd Jamil, Sazaly AbuBakar, Yulan Lin, Hai Yen Lee

**Affiliations:** 1Centre for Epidemiology and Evidence-Based Practice, Department of Social and Preventive Medicine, Faculty of Medicine, Universiti Malaya, Kuala Lumpur 50603, Malaysia; 2Department of Epidemiology and Health Statistics, School of Public Health, Fujian Medical University, Fuzhou 350122, China; 3Tropical Infectious Disease Research and Education Centre (TIDREC), Higher Institution Centre of Excellence (HiCOE), Universiti Malaya, Kuala Lumpur 50603, Malaysia

**Keywords:** dengue fever, indigenous community, knowledge, belief, prevention practices

## Abstract

Background: Dengue is prevalent among the indigenous community due to impoverished living conditions near the forest fringe areas and lack of health awareness. The study aims to determine the effect of a dengue awareness calendar on knowledge, belief, and practices (KBP) among the indigenous population. Method: A cross-sectional study was conducted in nine selected indigenous villages in Selangor, Malaysia. A dengue awareness calendar was distributed to the indigenous communities after pre-intervention. The KBP scores were compared between the pre-and post-intervention. Result: A total of 609 paired responses were obtained. Knowledge, perceived severity, cues to action, self-efficacy, and prevention practices were significantly improved after the intervention (*p* > 0.00). Participants with primary (Odds Ratio (OR) 2.627; 95% CI 1.338–5.160) and secondary level education (OR 2.263; 95% Cl 1.126–4.550) reported a high increment in practices score. High increments in dengue knowledge scores (OR 2.190; 95% CI 1.521–3.757, *p* < 0.00) were significantly more likely to report a high increment in practices score. Housewives (OR 0.535; 95% Cl 0.289–0.950), perceived severity (OR 0.349; 95% CI 0.184–0.662), and perceived susceptibility (OR 0.474; 95% CI 0.286–0.785) were significantly less likely to report an increment in prevention practices score. Conclusion: Findings inferred that the dengue awareness calendar significantly improved knowledge and practices. Our findings revealed the effectiveness of the dengue awareness calendar in dengue prevention among indigenous communities.

## 1. Introduction

Dengue fever is an alarming epidemic worldwide, and 3.6 billion people are at high risk of being infected with dengue fever [1]. Malaysia was ranked third in the number of cases due to the hyper-endemic outbreak of dengue [2]. The incidence and death of dengue in Malaysia have grown dramatically over the years. The highest number of dengue cases was reported in 2019, with 130,101 cases, and the highest dengue mortality cases were recorded in 2015, with 336 deaths. (Department of Statistics Malaysia, 2020).

The increasing world population, unplanned rapid expansion of urbanization, global warming, ineffective mosquito control methods, inadequate health care facilities, migration, globalization travel and trade, and an uneven climate are some of the key factors contributing to the increase in disease transmission, spreading to new territories and rural areas [3,4]. Past studies in Malaysia showed that dengue has spread from urban to rural areas, including forest fringe areas where most of the indigenous community resides [5,6]. Indeed, in 2014, a study revealed that seroprevalence cases were reported significantly higher in rural areas compared to urban areas [7]. The indigenous population, also known as *Orang Asli* (in the Malaysian language), is the minority in Malaysia and accounts for less than 1% of the total population in Malaysia. Indigenous people are classified into three main ethnolinguistic groups, namely Senoi, Proto-Malays, and the Negritos, each consisting of different dialectic subgroups and geographical locations [8].

The control of mosquito-borne viral infections is very challenging, especially in rural and forest fringe areas [7]. Primary prevention, such as vector control, is the key to overcoming dengue phenomena but is often inhibited due to low community support and involvement [9,10]. Early detection and prompt access to medical care are the prime factors in reducing fatalities in the absence of any specific treatment [11]. Community participation is vital for successful prevention and relies heavily on awareness, knowledge, and attitude about the disease, mode of transmission, and breeding sites [12]. Knowledge, attitude, and practices (KAP) research is often used in assessing the level of awareness and disease prevention practices [13]. Studies involving dengue and its prevention among the indigenous community are lacking in Malaysia. To date, only one study has been performed to assess the KAP of this community in Malaysia [14].

Health education plays an important role in providing adequate knowledge of the disease and its vector, as well as promoting behavioral changes such as following proper prevention practices to reduce the severity of cases [15,16]. Intervention targeting environmental cleanliness, vector control, and changing human behavior is important. This dengue awareness calendar was designed to assist the user in improving their dengue knowledge and health beliefs and subsequently performing the correct dengue prevention practices as their daily routine through vibrant infographic design and daily visual contact with the calendar. Our main objective is to evaluate the effectiveness of the dengue awareness calendar in enhancing dengue knowledge, health beliefs, and prevention practices among indigenous communities in Selangor, Malaysia. Moreover, this study aims to identify the factors that influence the increase in dengue prevention practices.

## 2. Materials and Methods

### 2.1. Sampling Frame

We conducted a cross-sectional study among the indigenous community in Selangor state in Peninsular Malaysia. The sample for this study was indigenous people originating and living in the selected village. According to the Department of Orang Asli Development (JAKOA), there are seven out of nine districts in Selangor where indigenous people are found. From these seven districts, two villages from each district where JAKOA was able to assist in accessing the indigenous community of the respective districts were selected using purposive sampling. In total, nine indigenous villages were selected based on (1) the accessibility of these villages by land transport, (2) permission granted by the head of the villages, and (3) a large number of populations in each village. Each household in the selected villages was approached, and only one person was surveyed. If there was more than one eligible person available in a household, one participant was selected randomly. If participants refused to be interviewed or if the resident of the house was not present, it was regarded as a non-response. Inclusion criteria for the study were (1) residents aged 18 and above, (2) willing to provide verbal informed consent, (3) able to understand and comprehend Bahasa Malaysia, and (4) willing to provide a telephone number. The sample size was calculated using the Daniel (1999) [17] sampling method equation: n = Z^2^* [p(1 − p)/d^2^]. Using a margin of error (d) of 0.05 (5%), with a 95% CI, chi-square value (Z) of 1.96, and 50% expected rate of dengue (p), the calculated sample size was 384. An extra 10% was added to the estimated sample size to account for potential missing values and invalid responses, and the minimum survey sample size was set to 422 participants.

### 2.2. Dengue Awareness Calendar

The dengue awareness calendar was provided in the local language (Malaysian language or Bahasa Malaysia) for the convenience of the indigenous community. The content of the dengue prevention education in the calendar was developed and validated by a panel of experts consisting of physicians and academicians who specialized in epidemiology, infectious disease, and microbiology. The calendar was designed with five important key messages (1) the knowledge of dengue vectors, (2) Aedes mosquitoes characteristics, (3) dengue transmission, (4) correct measures to eliminate the mosquito breeding sites, and (5) proper prevention of mosquito bites (Appendix A). The calendar was made in a form that could be hung anywhere inside the house so that the participants could continuously see the messages displayed on the calendar.

### 2.3. Research Instrument

The survey questionnaire of pre- and post-intervention consisted of five sections accessing (1) socio-demographic characteristics, (2) symptomatic dengue experience and environmental surroundings, (3) dengue-related knowledge, (4) health beliefs, and (5) self-reported prevention practices (Appendix A). Questions on knowledge about dengue transmission, symptoms, control, and treatment consisted of 6 sections (40 items), i. general information on dengue and the Aedes spp. mosquito (10 items), ii. transmission of dengue (9 items), iii. proper prevention of dengue (5 items), iv. signs and symptoms of dengue (14 items), and v. treatment and preventive measure (2 items). For each statement, the answer options were “yes”, “no”, or “don’t know”. The correct response was given a score of one, and incorrect or “don’t know” scored zero. The six items where the correct response is false were reverse coded.

The knowledge score was categorized based on the median split; therefore, in the pre-intervention, the knowledge score was categorized into two groups, low score (11–26) and high score (27–36); post-intervention, the low score (13–32) and high score (33–40). The differences between post- and pre-intervention dengue knowledge scores were also calculated. The increment was categorized into 0–6 and 7–17, with a higher range implying a higher increment.

The health belief model (HBM) has been applied extensively to study health beliefs that explain, predict, and influence behaviors [18]. In this study, belief questions based on several constructs of the HBM were used to evaluate the participants’ intentions and actions in dengue prevention, each ranging on a scale of 0–10. (i) Perceived severity refers to the seriousness of dengue; (ii) perceived susceptibility explains the vulnerability of being infected with dengue fever; (iii) perceived barriers assess the obstacles faced to prevent dengue (lack of community participation, lack of self-efficacy, and lack of preventive measures; (iv) cues to action evaluates the motivation to perform the dengue prevention practices (e.g., death, encouragement from a non-governmental organization (NGO), neighborhood infected with dengue, enlightenment from mass media, sudden fogging by authorities); (v) self-efficacy measures the confidence level of an individual engaging in protective practices. All items in the attitude question were summed to create a score with a higher score range indicating a higher level of positive attitude. Pre- and post-differences were categorized into (1) post-intervention score is the same or less than the pre-intervention score (Post ≤ pre) is regarded as no increment and (2) post-intervention score is more than the pre-intervention (Post > pre) is regarded as having increment in positive attitude.

Self-reported prevention practices of dengue consisted of 19 questions, i. prevention of mosquito breeding sites (9 items), ii. prevention of mosquito bites (7 items), and iii. prevention of dengue transmission (3 items). For each question, the response options were “never”, “rarely”, “sometimes”, and “often”, scored as 0, 1, 2, and 3, respectively. The possible total prevention scores ranged from 0 to 57 points, where higher scores implied a greater level of self-reported dengue prevention practices. The prevention score was categorized based on the median split; therefore, in the pre-intervention, the prevention score was categorized into two groups, (1) low prevention practices (score 10–25) and (2) high prevention practices (score 26–40); post-intervention, (1) the lower prevention practices (score 21–43) and (2) high prevention practices (score 44–57). The differences between post- and pre-intervention dengue prevention practices scores were also calculated. The increment was categorized into a lower increment of prevention practices (score 0–17) and a higher increment of prevention practices (score 17–34). The items for knowledge, perceived severity, perceived susceptibility, perceived barriers, cues to action, self-efficacy, and self-reported practice questions had reliability (Cronbach’s α) of 0.904, 0.817, 0.802, 0.885, 0.840, and 0.869, respectively. All the questions were developed in reference to previous literature [12] by the research team and validated by a panel of experts that consisted of physicians and academicians. The questionnaire was developed in English and translated into Bahasa Malaysia. The translated questionnaires were reviewed by an independent translator, and back translation was conducted on the primary translated version. The questionnaire was pilot tested for clarity on a total of 52 indigenous people randomly selected from the study population. The pre-intervention was performed face-to-face by a team of trained enumerators. The enumerators were trained to reduce interviewer-related errors by ensuring that all respondents were asked identically worded questions without unscripted commentary that could bias responses. Participation was voluntary, and the participants provided written informed consent before the start of the interview. Upon completion of the pre-intervention, enumerators explained, distributed, and encouraged the study participants to hang the dengue calendar. The study participants were also informed on the post-intervention questionnaire survey after six months using telephone interviews.

### 2.4. Statistical Analysis

Descriptive statistics were used to describe the proportion of knowledge, belief, and dengue prevention practices. The normality of total knowledge and self-reported prevention practices scores were checked using the Kolmogorov–Smirnov test. The comparison of scores pre- and post-intervention was performed using the Wilcoxon signed-rank test. Multivariable logistic regression for the outcome variable of self-reported dengue prevention practices score included demographic characteristics, experience, and environmental factors, increment in knowledge score, and differences in health belief. In the multivariable logistic regression analyses, all variables found to have a statistically significant association (two-tailed, *p*-value < 0.05) in the univariate analyses were entered into the model via the forced-entry method. The increase in prevention practices score was not normally distributed; therefore, multivariate logistic regression was used in the analysis. The dependent variable in multivariable logistic regression analysis was the increment in prevention practices scores that were categorized as a score of 17–34, representing 1, vs. a score of 0–16, representing 0. The independent variables were socio-demographic characteristics, increment in knowledge, and health beliefs scores. Odds ratios (OR), 95% confidence intervals (95%CI), and p-values were calculated for each independent variable. The model fit was assessed using the Hosmer–Lemeshow goodness of fit test. A p-value of less than 0.05 was considered statistically significant. All statistical analyses were performed with the Statistical Package for the Social Sciences Version 23.0 (SPSS; Chicago, IL, USA).

## 3. Results

### 3.1. Socio-Demographic Characteristics, Symptomatic Dengue Experience, and Surrounding Environment

The demographic characteristics of the 609 participants are shown in Table 1. A majority of the participants were aged between 31 to 50 years old (44.7%). The study had a slightly higher representation of female participants (59.1%, n = 360), participants with secondary level education attainment (47.8%, n = 291), and with an average monthly household income of less than Malaysian Ringgit (MYR) 1000 (60.8%, n = 289). Less than half (n = 291, 47.8%) of the participants attained secondary-level education and were housewives (39.7%). By tribe, the majority were Temuan (58.9%, n = 359). In the self-reported survey among 609 indigenous participants, only 4.8% (n = 29) had dengue fever.

### 3.2. Dengue Knowledge

Appendix A shows the correct responses to knowledge items. In general, the proportion of correct responses increased from pre- to post-intervention. The majority had a good knowledge of the mosquitoes, and over 90% of participants provided correct responses. Other increases in the knowledge dimension included protective measures against dengue infection and increased percent scores of dengue transmissions through blood (38.4%), dengue hemorrhagic symptoms (27.1%), and blood in urine (26.0%), Aedes mosquito do not live in places with lots of plants (24.6%), and Aedes mosquito egg contains dengue virus (24.5%). The least increased knowledge score in post-intervention was pain in the eyes (3.3%) as one of the symptoms of dengue.

### 3.3. Health Beliefs

In the health beliefs section, analysis of the perception of severity, susceptibility, barriers, cues to action, and self-efficacy are shown in Table 2. The pre-intervention revealed that the majority of the participants had a higher agreement score (6–10) for perceived severity (77.0%), perceived susceptibility (57.3%), perceived barriers (52.1%), and self-efficacy (85.9%). The proportion of the participants who had a higher agreement score (6–10) for all health belief items increased in the post-intervention assessment. The highest increment was observed for ‘cues to action’ (49.6% pre-intervention vs. 81.5% post-intervention).

### 3.4. Prevention Practices

Appendix A shows the self-reported dengue prevention practices by the participants in the study. Overall, increases in practice scores were observed for all items. The highest increment in the proportion of “often” practicing post-intervention was reported for preventing mosquitoes from biting a dengue patient (15.9% pre-intervention vs. 52.7% post-intervention), and the lowest was avoiding sexual intercourse with a spouse infected with dengue (5.4% pre-intervention vs. 15.3% post-intervention).

### 3.5. Overall Comparision of the Variables

Table 3 summarizes the median and interquartile range (IQR) of dengue knowledge score, health beliefs towards dengue agreement score, and dengue prevention practices scores pre- and post-intervention. Overall, all variable scores showed an increase from pre- to post-intervention. The median score of dengue knowledge increased significantly (*p* < 0.001) from pre-intervention (26.0) to post-intervention (32.0). The median score of dengue prevention practices increased significantly (*p* < 0.001) from pre-intervention (24.0) to post-intervention (43.0).

### 3.6. Factors Associated with the Increment of Dengue Prevention Practices Score

The multivariate logistic regression analysis (Table 1) indicates that the tribe ‘Temuan’ were significantly less likely to have a high increment in practice score (OR 0.444, 95%CI 0.254–0.777) compared to “other” tribes. Participants who are housewives were less likely (OR 0.535, 95%CI 0.289–0.950) to have a high increment in practices score compared to the “other” occupation category. Compared to participants with non-formal education, those with primary education level and secondary or above level education showed a high increment in practices score ((OR 2.627, 95%CI 1.338–5.160); OR 2.263, 95%CI 1.126–4.550, respectively).

Participants with a higher dengue knowledge score (7–17) were more likely (OR 2.390, 95%CI 1.521–3.757) to have a high increment in practices score. Participants with no increment in perceived severity (OR 0.349 95%CI 1.521–3.757) and no increment in perceived susceptibility (OR 0.474 95%CI 0.286–0.785) were less likely to have a higher increment in practices score.

## 4. Discussion

This study shows an overall increase in all knowledge, belief, and practices dimensions studied after awareness intervention using a calendar among the indigenous participants. For the indigenous community, multiple efforts have been in place to advance the lifestyle and well-being of the community [19]; however, there are still many efforts to be made to achieve better literacy about dengue prevention in the community. A previous study [14] reported that skilled workers in the indigenous community who have higher educational levels reported higher knowledge scores compared to those who were unemployed, including housewives; this is in concordance with our current findings. A study in Yogyakarta, Indonesia, reported that housewives are an important target group for dengue prevention practices [20]. In the indigenous communities, most housewives were less likely to receive high education; therefore, our findings suggest that awareness campaigns should place a higher emphasis on housewives. Furthermore, housewives spend most of their time at home, and they are also the prime person that carries out household control of mosquito breeding sites.

In this study, the awareness of dengue among this indigenous community is considerably high in the knowledge of the dengue virus and the role of mosquitoes in the transmission of dengue, as similarly found in previous studies in Malaysia [21]. However, a lower increment in the knowledge scores was observed in areas of preventive measures and signs of dengue hemorrhagic fever (DHF), suggesting a gap in the knowledge, findings which provide insights into areas of improvement for future community-based intervention programs. On a positive note, this study found an overall improvement in identifying DHF signs, perception of the severity of dengue, and prevention practices. Perception of severity played an important role in behavioral prevention, as it was evident that increased perception of severity may promote mosquito control practices in Zika virus prevention [22].

Findings about the perception of severity and perception of susceptibility to dengue showed that, although many were concerned about the likelihood of dengue infection, less than half of the participants viewed themselves as at low risk of becoming infected with dengue. This was also reported in another study [23], where the study participants expressed fear about dengue; however, the perception of vulnerability was low. A study by Becker (1974) quoted that “one’s intention to self-care is influenced by his or her perception of vulnerability and the severity of disease outcomes”. This indicates the importance of dengue awareness to enhance the risk perception among the indigenous community, as high-risk perception translates into protective behaviors. People with a high perception of self-efficacy in taking measures to prevent dengue reported greater commitment to engaging in activities [21]. A similar observation was reported among the rural population in Kuala Kangsar, where respondents believed that the responsibility of Aedes control lies within the community, and they would support the health authorities in any campaigns or activities that aimed to eradicate dengue [24]. This indicates that dengue awareness is useful in enhancing dengue prevention practices.

With regard to dengue practices, over half reported an increment in score from pre- post-intervention, which shows the dengue prevention education message in the calendar may have positively encouraged the study participants to enhance their engagement in mosquito prevention practices. The high level of prevention practices found in this study was related to daily practices for controlling mosquito nuisance, which was similarly reported in other studies [23]. The majority of participants took action by clearing mosquito breeding sites, as reported in post-intervention, which was also similarly reported in Saudi Arabia [25]. The improvement in prevention practices, such as proper disposal of household garbage, covering all water containers, and changing stored water to eliminate the mosquito-breeding site, was observed in this study. This shows that the message displayed on the dengue awareness measures to avoid mosquito breeding grounds in the calendar shows a positive influence on the community. It is also worth noting that in various studies, the prevention of mosquito bites usually involves the use of mosquito coils to repel the mosquitoes from biting, as also reported in India, Malaysia, the Philippines, and Pakistan [24,26,27,28]. The reason for the common use of mosquito coils could be due to advertising in the media. Our study found an increase in the use of aerosol insecticides post-intervention. This is similar to a study conducted in Puerto Rico, where exposure to posters increased the indoor use of aerosol insecticides [29]. However, it is important to note that preventive practices are subject to affordability; people in the Philippines and the rural and slum communities in India reportedly did not use any insecticidal sprays as they considered this prevention an expensive practice considering most of them have limited financial capabilities [27,28].

Fewer respondents use mosquito repellent and wear bright-colored clothing to avoid mosquito bites in post-intervention, similar to a community-based study in Brazil [30]. A simple practice such as a change in the color of the clothes can potentially help in reducing the risk of mosquito bites, as mosquitoes are attracted to darker areas and colors, which should be made known to the community. The present study also highlights the importance of educational interventions in many forms, including daily broadcasts, dengue campaigns, and group education, as reported in [31]. A noteworthy finding from the multivariate analysis shows the importance of HBM constructs in dengue prevention. It was found that higher perceived severity and susceptibility to dengue were associated with higher dengue prevention practices. Thus, low severity and perceptions of risk may reduce motivation to take action against mosquito prevention. Therefore, interventions should enhance the perception of the serious threat of dengue, and adding testimonials from people with previous dengue exposure or lost family member due to dengue would be useful. Lastly, there is a significant positive association between an increment in knowledge score and an increase in mosquito prevention practices, which aligned with previous studies [32,33]. This indicates that low knowledge of dengue transmission, prevention, and treatment leads to poorer prevention practices against dengue.

There are a few limitations seen in this study. Firstly, the selection of indigenous villages was based on JAKOA approval that relied on accessibility by land transport and approval by the Head of the village; therefore, the findings may not be representative of all indigenous people in general. Secondly, the cross-sectional design used could not infer a causal relationship. Furthermore, other barriers in prevention practices are not able to be captured in qualitative study design. A future study using qualitative methods, such as focus group discussion, would be advantageous to gauge an in-depth understanding of prevention barriers in this community. Thirdly, the self-reporting data may be subjected to reporting bias towards socially desirable responses.

## 5. Conclusions

The novelty of the study was shown by the improved knowledge, belief, and practices of the indigenous community after the distribution of the dengue awareness calendar, which is a new mode of awareness intervention delivery. The calendar’s content encouraged indigenous communities to perform better prevention activities. The dengue awareness calendar is a good substitute for other promotional materials as people may likely display the calendar at home; hence, it may act as a constant reminder for them to perform mosquito prevention practices. Moreover, it is a relatively low cost, and the infographics in the calendar highlight steps of dengue prevention practices in an easy-to-understand way. The demographic disparities in prevention practices found in this study provide insights for intervention tailored to specific demographic groups in the indigenous communities. Future intervention programs should consider literacy-related barriers in their implementation. The findings provide important insights for policymakers to consider using this approach in future dengue prevention programs.

## Figures and Tables

**Table 1 healthcare-11-00637-t001:** Association between socio-demographic characteristics, increment in knowledge score, and differences in health beliefs with increment in prevention practices score (N = 609).

Details	n (%)	Increment in Prevention Practices Score			Multivariate Logistics Regression for 17–34 vs. 0–16
Socio-Demographic Data		0–16 (n = 311)	17–34 (n = 298)	*p*-Value	
Age group (years old)					
18–30	215 (35.3)	112 (52.1)	103 (47.9)		
31–50	272 (44.7)	138 (50.7)	134 (49.3)	0.924	
>50	122 (20.0)	61 (50.0)	61 (50.0)		
Gender					
Male	249 (40.9)	108 (43.4)	141 (56.6)		1.357 (0.790–2.329)
Female	360 (59.1)	203 (56.4)	157 (43.6)	0.002	Reference
Tribe (N = 615)					
Temuan	359 (58.9)	202 (56.3)	157 (43.7)		0.444 (0.254–0.777) *
Mahmeri	127 (20.9)	53 (41.7)	74 (58.3)	0.007	0.608 (0.312–1.184)
Others	123 (20.2)	56 (45.5)	67 (54.5)		Reference
Education					
No formal	82 (13.5)	58 (70.7)	24 (29.3)		Reference
Primary level	236 (38.8)	108 (45.8)	128 (54.2)	0.00	2.627 (1.338–5.160) *
Secondary and above level	291 (47.8)	145 (49.8)	146 (50.2)		2.263 (1.126–4.550) *
Occupation					
Manual worker	203 (33.3)	88 (43.3)	116 (56.7)		1.113 (0.663–1.870)
Housewife	242 (39.7)	144 (59.5)	98 (40.5)	0.002	0.535 (0.289–0.950) *
Others	164 (26.9)	79 (48.2)	85 (51.8)		Reference
Monthly income (MYR) [N = 475]					
≤1000	289 (60.8)	161 (55.7)	126 (44.3)	0.019	Reference
>1000	186 (39.2)	83 (44.6)	103 (55.4)		1.142 (0.706–1.851)
Symptomatic Dengue Experiences					
Yes	29 (4.8)	18 (62.1)	11 (37.9)	0.257	
No	580 (95.2)	293 (50.5)	287 (49.5)		
Environmental Factors					
Density of plants or vegetation					
None/Low	162 (26.6)	89 (54.9)	73 (45.1)		
Moderate	231 (37.9)	118 (51.1)	113 (48.9)	0.426	
A lot	216 (35.6)	104 (48.1)	112 (51.9)		
Abundance of mosquitoes in neighborhood					
None/Low	224 (36.8)	117 (52.2)	107 (47.8)		
Moderate	254 (41.7)	127 (50.0)	127 (50.0)	0.888	
Severe	131 (21.5)	67 (51.1)	64 (48.9)		
Frequency of mosquito fogging					
None/Rarely	448 (73.6)	225 (50.2)	223 (49.8)		
Occasionally /Often	161 (26.4)	86 (53.4)	75 (46.6)	0.520	
Increment in Knowledge Score					
0–6	367 (60.1)	220 (59.9)	147 (40.1)	0.00	Reference
7–17	242 (39.9)	91 (37.6)	151 (62.4)		2.390 (1.521–3.757) ***
Differences in Health Beliefs					
Perceived Severity					
Have increment	91 (14.9)	29 (31.9)	62 (68.1)	0.00	Reference
No increment	518 (85.1)	282 (54.4)	236 (45.6)		0.349 (0.184–0.662) **
Perceived Susceptibility					
Have increment	114 (18.7)	44 (38.6)	70 (61.4)	0.004	Reference
No increment	495 (81.3)	267 (53.9)	228 (46.1)		0.474 (0.286–0.785) *
Perceived Barriers					
Have increment	119 (19.5)	53 (44.5)	66 (55.5)	0.125	
No increment	490 (80.5)	258 (52.7)	232 (47.3)		
Cues to Action					
Have increment	205 (33.7)	88 (42.9)	117 (57.1)	0.005	Reference
No increment	404 (66.3)	223 (55.2)	181 (44.8)		0.694 (0.448–1.076)
Self-Efficacy					
Have increment	63 (10.3)	27 (42.9)	36 (57.1)	0.185	
No increment	546 (89.7)	284 (52.0)	262 (48.0)		

* *p* < 0.05, ** *p* < 0.01, *** *p* < 0.001. Hosmer and Lemeshow test, c2(8) = 10.584, *p* = 0.226; Cox and Snell R^2^ = 0.153; Nagelkerke R^2^ = 0.205. OR, odds ratio; CI, confidence interval; –, not applicable in the multivariate analysis.

**Table 2 healthcare-11-00637-t002:** The frequency and percentage of agreement score responses on the health beliefs in pre- and post-intervention [N = 609].

Details	Frequency, n (%)	
(D) Health Belief Model	Pre-intervention	Post-intervention
Perceived Severity		
Seriousness of dengue		
0–5	140 (23.0)	61 (10.0)
6–10	469 (77.0)	548 (90.0)
Perceived Susceptibility		
Worried about the likelihood of getting infected with dengue		
0–5	260 (42.7)	249 (40.9)
6–10	349 (57.3)	360 (59.1)
Perceived Barriers		
Concern about the lack of community participation, lack of self-efficacy, and lack of preventive measure		
0–5	292 (47.9)	243 (39.9)
6–10	317 (52.1)	366 (60.1)
Cues to Action		
Motivation to prevent dengue, e.g., death, encouragement from NGO, neighborhood infected with dengue, enlightenment from mass media, sudden fogging by authorities		
0–5	307 (50.4)	112 (18.4)
6–10	302 (49.6)	497 (81.5)
Self-Efficacy		
Confidence level to prevent dengue		
0–5	86 (14.1)	61 (10.0)
6–10	523 (85.9)	548 (90.0)

**Table 3 healthcare-11-00637-t003:** Wilcoxon Signed Rank test for dengue knowledge score, level of health beliefs, and prevention score in the pre- and post-intervention.

Variable	Pre-Test Median (IQR)	Post-Test Median (IQR)	Z-Value	*p*-Value
Knowledge score	26.0 [19.0–30.0]	32.0 [26.0–36.0]	21.20	*p* < 0.001
Health Beliefs				
Perceived severity	10.0 [8.0–10.0]	10.0 [9.0–0.0]	4.45	*p* < 0.001
Perceived susceptibility	6.0 [4.0–8.0]	6.0 [4.0–8.0]	0.28	0.779
Perceived barriers	6.0 [3.0–8.0]	6.0 [4.0–8.0]	1.29	0.199
Cues to action	5.0 [4.0–9.0]	6.0 [4.0–8.0]	3.48	*p* < 0.001
Self-efficacy	8.0 [6.0–10.0]	8.0 [7.0–10.0]	3.54	*p* < 0.001
Practices score	24.0 [21.0–29.0]	43.0 [36.0–48.0]	21.22	*p* < 0.001

## Data Availability

Data sharing is not applicable to this article, as no data sets were generated or analyzed during the current study. All data given are available in the original articles.

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
