# Peer review of "Effectiveness of Dengue Awareness Calendar on Indigenous Population: Impact on Knowledge, Belief and Practice"

_healthcare, 2023, doi:10.3390/healthcare11050637_

Round 1

Reviewer 1 Report

The paper entitled “Effectiveness of Dengue Awareness Calendar among Indigenous
Population: Impact on knowledge, belief and practice
” is interesting, informative, and could be useful for policymakers in responding to the fact that the dengue awareness calendar significantly improved knowledge and practices in the indigenous community. I have the following comments and suggestions as below:

  1. The effectiveness of (KBP) was measured using an awareness calendar during the pre-and post-intervention period among the indigenous population in nine villages. The highest increment was observed after the intervention. Do you think it is difficult to quantify belief? Could you please explain how the health belief model can help with evaluation? 
  2. It will be useful if one can use a dengue awareness calendar in the endemic region. How practical it would be to use such a method in rural communities?
  3. Do the authors explain the advantage of a cross-sectional study over a mixed method in the current study?
  4. What about the other methods for accessing the effectiveness of interventions, including focus group discussion (FGD)? It may add value to see the impact of the study.
  5. Given the limited scope of the study, how do the authors intend to present their findings to policymakers?
  6. How do you link socioeconomic status and gender? Some of the already known factors, like level of education, monthly income, etc., have a positive impact on the awareness of dengue.
  7. The authors mentioned that this study was conducted in a single indigenous population and may have difficulties generalizing the findings. Do the authors have plans to replicate this study in another province or other indigenous community to influence policymakers?

Author Response

Please find our detailed responses in the reply letter.

Reviewer 2 Report

Effectiveness of Dengue awareness calendar among indigenous population: impact on knowledge, belief and practice.

Li Ping Wong et al.

The authors in this report designed a dengue awareness calendar and distributed among the indigenous population in selected villages in Malaysia. The aim of the study was to improve/monitor the participant’s knowledge on dengue infection and health beliefs, so that correct dengue prevention measures are practiced in their daily routines. The study is reasonable and needs to be published. I have few minor comments. As the authors indicated that the role of housewives in disease prevention is critical. Therefore, educating housewives in the indigenous community will certainly improve the prevention practices. However, there were several male respondents in the study, and the education levels among the family members are not known, which may influence the use of correct prevention practices. This limitation should be mentioned in the discussion. Manuscript needs minor proofreading.  

Minor comments:

Define abbreviations, e.g., KBP in the abstract, OR, Equation used to calculate the sample size, pre- and post-intervention

Line 102 should be microbiology

Table 1. Dengue experiences; I would say symptomatic dengue experiences, as most DENV-infected people have asymptomatic infections.

Author Response

(The authors gave the same response as above.)

Reviewer 3 Report

The study aims to evaluate the effectiveness dengue awareness calendar in enhancing dengue knowledge, health beliefs, and prevention practices among Indigenous communities in Selangor, Malaysia as well as to identify the factors that influence the increase in dengue prevention practices. Overall, this is an important study, but some points need to be improved and clarified.

Lines 92-93: “The sample size was calculated using the equation: = Z2 p(1-p)/d2. Using a margin of error of 0.05 …”

I would suggest explaining the equation. What are Z2, p, and d2?

Line 105, Figure S1, the Dengue Awareness Calendar:

I would suggest increasing the resolution of the Figure. Some contents are unreadable.

Line 158: “The questionnaire was pilot tested on the random Indigenous population.”

I would suggest providing the reason for pilot testing and what the results are

Line 167: Under Statistical Analysis, I would suggest providing the Logistic Regression Model and providing more explanation regarding the formula. Please state clearly the dependent variable and the independent variables. Why Logistic Regression Model is appropriate?

Line 187: Please explain MYR! Please also revise the punctuation in this line!

Line 189: “By Tribe, the majority were Temuan (58.9%, 359).”

Please be consistent! By Tribe, the majority were Temuan (58.9%, n=359).

Line 198: Figure S3. Please increase the resolution of Figure S3!

Line 198: Figure S3 shows the correct responses of knowledge items. What do you mean by “correct responses”?

Line 221: Please revise the punctuation in this line!

Line 2447: “Participants with higher dengue knowledge score (7-17) were more likely (OR 2.190…”

This explanation is based on Table 1, but I could not see the OR 2.190 in the Table. Please explain it!

Author Response

(The authors gave the same response as above.)
